# Development of a PCR Assay for the Detection of *Legionella micdadei* in the Environment

**DOI:** 10.3390/idr17050131

**Published:** 2025-10-17

**Authors:** William N. Bélanger, Martine Bastien, Eve Bérubé, Martin Gagnon, Yesmine G. Sahnoun, Valérie Dancause, Karel Boissinot, Cindy Lalancette, Christian Riel-Roberge, Marieve Jacob-Wagner, Sylvie Trottier, Damien Biot-Pelletier, Annie Ruest, Isabelle Tétreault, Mathieu Thériault, Sandra Isabel

**Affiliations:** 1Département de Biochimie, Microbiologie et Bio-Informatique, Faculté Des Sciences et de Génie, Université Laval, Quebec City, QC G1V 0A6, Canada; william.belanger@crchudequebec.ulaval.ca; 2Axe Maladies Infectieuses et Immunitaires, Centre de Recherche du CHU de Québec-Université Laval, Quebec City, QC G1V 4G2, Canada; martine.bastien@crchudequebec.ulaval.ca (M.B.); eve.berube@crchudequebec.ulaval.ca (E.B.); martin.gagnon@crchudequebec.ulaval.ca (M.G.); yesmine.sahnoun@crchudequebec.ulaval.ca (Y.G.S.); sylvie.trottier@crchudequebec.ulaval.ca (S.T.); mathieu.theriault@crchudequebec.ulaval.ca (M.T.); 3Centre de Recherche en Infectiologie, Université Laval, Quebec City, QC G1V 4G2, Canada; 4Programme de Prévention et de Contrôle des Infections, CHU de Québec-Université Laval, Quebec City, QC G1R 2J6, Canada; valerie.dancause@chudequebec.ca; 5Département de Microbiologie et D’infectiologie, CHU de Québec-Université Laval, Quebec City, QC G1S 4L8, Canada; karel.boissinot@chudequebec.ca (K.B.); marieve.jacob-wagner@chudequebec.ca (M.J.-W.); annie.ruest.med@ssss.gouv.qc.ca (A.R.); tetrault.isabelle.med@ssss.gouv.qc.ca (I.T.); 6Laboratoire de Santé Publique du Québec, Sainte-Anne-de-Bellevue, QC H9X 3R5, Canada; cindy.lalancette@inspq.qc.ca; 7Direction de Santé Publique, Centre Intégré Universitaire de Santé et de Services Sociaux (CIUSSS) de la Capitale-Nationale, Quebec City, QC G1E 7G9, Canada; christian.riel-roberge.ciussscn@ssss.gouv.qc.ca; 8Département de Microbiologie-Infectiologie et D’immunologie, Faculté de Médecine, Université Laval, Quebec City, QC G1V 0A6, Canada; 9Centre D’expertise en Analyse Environnementale du Québec (CEAEQ), Gouvernement du Québec, Quebec City, QC G1P 3W8, Canada; damien.biot-pelletier@environnement.gouv.qc.ca; 10Département de Pédiatrie, Faculté de Médecine, Université Laval, Quebec City, QC G1V 0A6, Canada

**Keywords:** *Legionella micdadei*, PCR assay, 23S–5S intergenic spacer, environmental detection

## Abstract

Background/Objectives: *Legionella micdadei* is a clinically significant species within the *Legionella* genus, requiring accurate detection methods, surveillance, and precise clinical diagnosis. Our objective was to develop a sensitive polymerase chain reaction (PCR) assay specific for *L. micdadei* to detect its presence in environmental specimens. Methods: We targeted the 23S–5S intergenic spacer region, which can differentiate *Legionella* spp. We tested the detection of *L. micdadei* with 20 strains and determined the limit of detection with 2 strains. We verified assay specificity with 17 strains of other *Legionella* spp., 62 strains of other bacterial and fungal genera, and three human DNA specimens. We evaluated intra- and inter-run precision. We tested 15 environmental specimens (water, swabs of water faucets, mulch, and soil) by PCR. Results: The PCR assay demonstrated 100% analytical specificity (no cross-reactivity with non-targeted species), 100% inclusivity (detection of all *L. micdadei* strains), and high precision, with a coefficient of variation ≤ 2% across replicates. The limit of detection was estimated at 5 genomic DNA copies per reaction. We detected *L. micdadei* in environmental specimens. Conclusions: This PCR assay enables accurate detection of *L. micdadei* and is not subject to competition with other *Legionella* spp., thereby addressing limitations of current broad-spectrum *Legionella* approaches. The evaluation supports its application in environmental detection for surveillance.

## 1. Introduction

*Legionella* spp. are Gram-negative mesophilic bacteria that are ubiquitous in freshwater ecosystems and soils [1]. Of the more than 60 species described, about half have been reported to infect humans [2]. These pathogenic bacteria are responsible for Legionnaires’ disease (LD) and Pontiac fever, both acquired through the inhalation of contaminated aerosols [3]. They can colonize water systems, including cooling towers, hot water tanks, and water distribution systems, posing significant public health risks [3]. In addition, certain *Legionella* spp. have also been detected in soils, potting mixes, and composts. These can serve as additional environmental reservoirs and sources of infection, particularly for individuals exposed to contaminated dust or aerosols during gardening or agricultural work [4,5,6]. From 2000 to 2019, the national incidence rate of Legionnaires’ disease in the United States increased 6.5-fold, from 0.42 to 2.71 cases per 100,000 persons [2]. While *L. pneumophila* causes approximately 90% of these infections, other species also contribute to the burden of disease [1]. Of these, the most frequently isolated from infected patients are *L. bozemanae*, *L. dumoffii*, *L. longbeachae*, and *L. micdadei* [7]. Based on 2018–2019 data from the Supplemental Legionnaires’ Disease Surveillance System (SLDSS), *L. micdadei* accounted for 1.3% of all clinical *Legionella* spp. cases diagnosed in the United States [2]. While *L. pneumophila* typically causes lobar consolidation or patchy pulmonary infiltrates, *L. micdadei* more frequently causes nodular lung lesions that may cavitate or enlarge over time [8,9]. This radiological pattern can closely mimic invasive fungal infections, potentially misleading clinicians toward a fungal diagnosis and prompt empiric antifungal therapy before the correct etiology is identified [10,11,12].

Standard detection methods currently rely on culture using buffered charcoal yeast extract (BCYE) agar media. Though specific for the *Legionella* genus, certain media formulations favor the rapid growth of *L. pneumophila*, while species like *L. micdadei* may require 1–2 weeks to form visible colonies due to their inherently slower growth rates, or fail to grow entirely [13]. This delay can hinder a timely response in water and cooling tower monitoring, when rapid intervention is critical to prevent and contain outbreaks. To improve selectivity, antibiotic supplements are added to BCYE agar to suppress bacteria and fungi other than *Legionella* spp. Some media, such as BCYE-MWY and BCYE-GVPC, support *L. micdadei* growth, whereas BCYE-BMPA inhibits it due to the presence of cefamandole [14]. Clinical laboratories usually use two to three culture media in parallel to detect a broader diversity of *Legionella* spp. However, this approach does not always include a medium suited for *L. micdadei*, which may lead to underdetection of this species [15]. Finally, *Legionella* spp. can be found co-occurring in environmental and clinical specimens, which can make their epidemiological correlation particularly complex [16,17,18].

Polymerase chain reaction (PCR)-based methods address these limitations by targeting conserved *Legionella* spp. DNA sequences, delivering results within 6 to 8 h [19,20,21]. They also detect low bacterial loads that can be missed by culture and allow for efficient monitoring of water systems [19,22]. While PCR cannot distinguish viable from non-viable bacteria, its speed and sensitivity support water and cooling tower monitoring, enable the detection of viable but nonculturable (VBNC) bacteria, and identify *Legionella* spp. contained within protozoa [22,23,24]. Multiple PCR-based methods have been developed for the detection and identification of *Legionella* spp., including both genus-level and species-specific assays targeting genetic regions such as 16S rRNA, *mip*, and various intergenic spacer regions [13,25,26,27,28,29,30]. While these approaches have improved diagnostic capabilities, the detection of *Legionella* spp. like *L. micdadei* remains challenging, especially in environmental specimens where specificity and sensitivity may be compromised by potential inhibitors and competing bacterial flora [30,31]. Applications that can discriminate between closely related *Legionella* spp. with high specificity in environmental matrices are limited. In this study, we present a PCR assay that specifically detects *L. micdadei* [25].

## 2. Materials and Methods

### 2.1. Bacterial and Fungal Strains

Twenty *L. micdadei* strains (Table 1), 17 strains from other *Legionella* spp., and 62 bacterial and fungal strains from other genera (Appendix A) were obtained from the American Type Culture Collection (Manassas, VA, USA) (ATCC), the Collection de Culture du Centre de recherche en Infectiologie (Québec, QC, Canada) (CCRI), and the Laboratoire de Santé Publique du Québec (Québec, QC, Canada) (L00 and ID). All *Legionella* spp. strains were grown at 35–40 °C under aerobic conditions on BCYE for 72 to 96 h. Other bacterial and yeast strains were cultured on tryptic soy agar with 5% sheep blood or Sabouraud dextrose agar for 24–48 h. Filamentous fungi were grown on potato dextrose agar for 7–21 days.

### 2.2. DNA Isolation

Bacterial DNA (both purified and crude genomic DNA (gDNA)) was isolated from cultures using one of the following methods: eMAG (Cat. No. 418591, bioMérieux, Marcy-l’Étoile, France), BioSprint 15 DNA Blood Kit (Cat. No. 940014, Qiagen, Mississauga, ON, Canada) automated with a KingFisher system (Cat. No. 5400610, Thermo Fisher Scientific, Waltham, MA, USA) [32], or crude lysate prepared using a standardized glass bead-beating lysis technique as previously described [33]. DNA concentration was measured using a NanoDrop spectrophotometer (Cat. No. ND-ONE-W, Thermo Fisher Scientific), a Qubit fluorometer (Cat. No. Q33226, Thermo Fisher Scientific), or a BioTek Synergy H1 (Cat. No. SYS-BT-SYNH1, Agilent, Santa Clara, CA, USA).

### 2.3. Primer and Probe Design

We targeted the three genome copies of 23S-5S rRNA intergenic spacers of *L. micdadei*, thereby increasing the number of detectable templates and improving assay sensitivity [34]. This spacer exhibits highly conserved extremities across the genus, while its central section varies between species, enabling species-specific discrimination. We retrieved sequences from GenBank and aligned them using MAFFT (version 7.520, RIMD, Osaka, Japan): ten sequences from four *L. micdadei* strains and 504 sequences from 22 other *Legionella* spp. (*L. anisa* (*n* = 9), *L. birminghamensis* (*n* = 1), *L. bozemanae* (*n* = 2), *L. cardiaca* (*n* = 1), *L. cherrii* (*n* = 4), *L. cincinnatiensis* (*n* = 1), *L. clemsonensis* (*n* = 1), *L. dumoffii* (*n* = 1), *L. feeleii* (*n* = 2), *L. gormanii* (*n* = 1), *L. hackeliae* (*n* = 5), *L. jordanis* (*n* = 5), *L. lansingensis* (*n* = 5), *L. longbeachae* (*n* = 30), *L. maceachernii* (*n* = 2), *L. oakridgensis* (*n* = 7), *L. parisiensis* (*n* = 1), *L. pneumophila* (*n* = 408), *L. rubrilucens* (*n* = 2), *L. sainthelensi* (*n* = 14), *L. tucsonensis* (*n* = 1), *L. wadsworthii* (*n* = 1)). We designed a forward primer, which we combined with the reverse primer and probe from Cross et al. (2016) to obtain specific amplification of *L. micdadei* (Table 2) [25]. The fluorophore tag of the probe was modified to CalRed610-BHQ2 to suit our detection system. Additionally, we designed two primers and a probe targeting *B. subtilis* strain CCRI-21428 as an internal control.

To assess inclusivity and specificity in silico, we analyzed primer and probe sequence cross-reactivity using Primer-BLAST (accessed August 2024, NCBI, Bethesda, MD, USA), querying the GenBank nucleotide collection (core_nt, National Center for Biotechnology Information, Bethesda, MD, USA). The search included all available complete and partial genome sequences of *L. micdadei* as well as sequences from 51 non-target species, selected based on phylogenetic relatedness and/or their presence in respiratory microbiota.

### 2.4. PCR Conditions

The PCR mixture (total 20 µL) contained four primers and two probes (Table 2), X FastStart Essential DNA Probes Master Mix (Roche Life Science, Basel, Switzerland), gDNA of *B. subtilis* (1000 genome copies) as an internal control, and 5 µL of specimens (Appendix A). PCR amplification was performed on a CFX96 thermal cycler (Bio-Rad, Hercules, CA, USA) as follows: 95 °C for 10 min, followed by 45 cycles of 95 °C for 10 s (ramp rate 1 °C/s) and 60 °C for 30 s, and finalized with a hold at 25 °C for 5 s. Each PCR was performed in duplicate (unless otherwise specified).

### 2.5. In Vitro Performance Characteristics of the PCR Assay

Analytical specificity was evaluated using purified gDNA and crude lysates from 17 *Legionella* strains other than *L. micdadei*, 56 other bacterial strains, six fungal strains, and three human gDNA (Appendix A) [35]. The inclusivity of the test for the target species was evaluated using 20 *L. micdadei* strains. Binomial 95% confidence intervals for analytical specificity and inclusivity were calculated using the exact Clopper–Pearson interval method on RStudio (v2025.05.1+513; Posit PBC, Boston, MA, USA) [36].

The limit of detection (LOD) was defined as the lowest genome copy number at which amplification was detected in at least 95% of replicates [37]. LOD evaluation involved testing two *L. micdadei* strains in eight replicates across five serial dilutions (20, 10, 5, 2, and 1 genome copies/reaction; Table 3) [37].

PCR efficiency and linearity were assessed using duplicate reactions of *L. micdadei* gDNA (10–10^5^ genome copies per reaction; Figure 1). To evaluate the precision of the PCR assay, we tested two strains of *L. micdadei* (Table 4). Mean Ct values and standard deviations were calculated for each strain, and the coefficient of variation (CV) was calculated for both repeatability (intra-assay) and reproducibility (inter-assay). Competition evaluation for the internal control. We evaluated the competition between *B. subtilis* internal control and *L. micdadei* detection. We mixed gDNA from *B. subtilis* at concentrations of 1000, 5000, 10,000, 50,000, and 220,000 genome copies with 10 genome copies of *L. micdadei* gDNA per reaction.

### 2.6. Competition Evaluation from Other Legionella spp.

To assess interference from other *Legionella* spp., purified gDNA or crude lysates from multiple *Legionella* spp. were tested in duplicates. Each reaction contained a fixed 10 copies of *L. micdadei* gDNA and one of 17 other *Legionella* spp. strains, introduced as either purified gDNA (1 ng) or crude lysate (1:100 dilution).

### 2.7. Environmental Specimen Preparation

One-liter tap water specimens were collected from hospital faucets in sterile bottles and processed on the same day, following ISO 12869:2019 guidelines. However, sodium thiosulfate was not added to the bottles to neutralize chlorine. Water specimens were filtered using a 0.4 µm membrane (PALL, Port Washington, NY, USA). The filters were folded and placed in a 2 mL plastic tube (Sarstedt, Nümbrecht, Germany). Sterile molecular-grade water (0.6 mL) was added to each tube. The tubes were vortexed three times for 30 s. Swabs of water faucets were collected using Roche Cobas flocked swab (Roche Diagnostics, Laval, QC, Canada). The swab was broken in a 15 mL conical plastic tube (Sarstedt, Nümbrecht, Germany) containing 1 mL of sterile molecular-grade water. The tubes were vortexed three times for 30 s. A volume of 50 µL of the suspensions was plated onto BCYE-MWY agar for culture, and 300 µL were transferred to 1.5 mL conical tubes with standardized glass beads for crude lysate DNA extraction [33]. The suspensions were centrifuged at 15,800× *g* for 5 min. The supernatants were discarded and the resulting pellets were resuspended in 50 uL of 5× TE buffer, (50 mM Tris-HCl, 5 mM disodium EDTA, pH 8.0; Thermo Fisher Scientific, Waltham, MA, USA), followed by 5 min bead-beating lysis and a 2 min heating step at 95 °C [33].

For gardening soil specimens, approximately 100 mg were suspended in 5 mL of sterile molecular-grade water and vortexed for 5 min. Specimens were then left to settle for 30 min. The clarified supernatants were collected and centrifuged at 500× *g* for 2 min to remove large debris. The resulting supernatants were filtered through a 5 µm syringe filter, then centrifuged at 10,000× *g* for 15 min to concentrate bacterial cells. After discarding the supernatant, the pellets were resuspended in 200 µL of sterile molecular-grade water by vortexing. 100 µL were then transferred into a glass-bead lysis tube, mixed with 900 µL of 1× TE buffer, and centrifuged at 21,000× *g* for 5 min. The supernatants were discarded. The resulting pellets were processed by glass-bead crude lysate DNA extraction as described above.

For dust and wood mulch mixed with soil, DNA was extracted and purified using the DNeasy PowerSoil Pro Kit (Cat. No. 47014, Qiagen, Hilden, Germany). Specimens were homogenized by manual mixing, then 250 mg were transferred to bead-beating tubes. Lysis and purification were then performed following the manufacturer’s instructions.

## 3. Results

### 3.1. In Silico Analysis of Primers and Probes

The in silico evaluation confirmed 100% identity for the primer and probe target regions across ten *L. micdadei* gene sequences available in GenBank, corresponding to four distinct strains (accession numbers: CP020615.1, Z24694.1, LN614830.1, and CP020614.1). Specificity analysis revealed between 7 and 17 mismatches in the forward primer compared to other *Legionella* spp., supporting assay specificity in silico.

### 3.2. In Vitro Performance Characteristics of the PCR Assay

We established the LOD at 5 genome copies of *L. micdadei* gDNA per reaction. With 5, 2, and 1 genome copies per reaction, the detection rates were 100% (32/32), 90.63% (29/32), and 68.75% (22/32), respectively (Table 3). The assay showed 100% inclusivity (20/20, 95% CI: 83.2–100%) for *L. micdadei* strains (Table 1). Analytical specificity was 100% (0/102 false positives; 95% CI: 96.5–100%), with no amplification detected in 17 strains of *Legionella* spp. other than *L. micdadei*, 56 bacterial strains from other genera, 6 fungal strains, and 3 human DNA specimens (Appendix A).

*L. micdadei* PCR showed linear amplification with an efficiency of 104% and a correlation coefficient (R^2^) of 0.998 (Figure 1A). We also tested the PCR precision on two strains of *L. micdadei.* Mean PCR cycle threshold (Ct) was similar, with average Ct values of 32.7 and 33.0, respectively. In all cases, the CV for repeatability and reproducibility was ≤2% (Figure 1B, Table 4).

### 3.3. Competition and Co-Amplification Analysis

We evaluated the potential for amplification competition within the assay by testing *L. micdadei* gDNA in the presence of high concentrations of internal control (*B. subtilis*) gDNA. The *L. micdadei* amplification exhibited no evidence of competition with the internal control gDNA, with a similar Ct value (31.38 ± 0.36) in all conditions (Figure 2A). Similarly, we observed no competition with other *Legionella* spp. during amplification, as indicated by comparable Ct values of 32.95 ± 0.25 for *L. micdadei* and 32.65 ± 0.27 for the internal control (*B. subtilis*) (Figure 2B).

### 3.4. Environmental Specimens

We evaluated the assay’s capacity to detect *L. micdadei* in environmental specimens: eight tap water, two swabs from water faucets, one dust, one wood mulch mixed with soil, and three gardening soil (Table 5). In parallel, tap water and swab specimens were also analyzed by culture. We detected *L. micdadei* by PCR in the wood mulch mixed with soil specimen and in one gardening soil specimen. However, we observed no amplification in any of the tap water, swab from water faucet, or dust specimens. Culture results identified the presence of *L. anisa*, *L. feeleii*, and *L. pneumophila* in five specimens. However, *L. micdadei* did not grow in cultures of environmental specimens tested.

## 4. Discussion

In this study, we developed a PCR assay targeting *L. micdadei*’s multi-copy 23S–5S rRNA intergenic region, including a *B. subtilis* internal control. The assay achieved high specificity for *L. micdadei*, low LOD, and high inclusivity with excellent repeatability and reproducibility. Even though detection of *L. micdadei* is rare in the environment, we found two sources (soil and mulch). Environmental matrices such as soil can harbor *L. micdadei* and serve as potential sources of exposure, as evidenced by our results [30]. Exposure to soil contaminated by *Legionella* spp. can occur through aerosolization during soil-disturbing activities and handling [37,38,39]. Thus, soil testing is important for outbreak investigation and surveillance.

The presence of multiple copies of the 23S–5S target region within the *L. micdadei* genome likely plays a role in efficiently detecting small amounts [34]. The present *L. micdadei*-specific PCR assay offers distinct advantages over previously published studies [25]. The Cross et al. assay detected *L. micdadei* at 10 genome copies per reaction with 50% positivity. Our assay demonstrated improved sensitivity with a lower LOD of 5 genome copies per reaction in ≥95% of replicates. Environmental matrices frequently contain substances like humic acids, which may co-extract with DNA and act as potent PCR inhibitors, interfering with nucleic acid amplification and potentially leading to false negatives or reduced assay sensitivity [38,39]. Thus, we incorporated an internal amplification control (*B. subtilis* DNA) to monitor that each reaction worked correctly. This internal control is especially important to detect false negatives when testing environmental specimens containing PCR inhibitors. Finally, our assay tolerates competition with more than 1000 fold excess concentration of other *Legionella* spp. strains [34]. In addition to these technical strengths, the species-specific PCR assay for *L. micdadei* offers operational advantages for public health and environmental monitoring. The rapid turnaround time (3–4 h) and low LOD facilitate *L. micdadei* detection and support surveillance of water and soil sources, potentially enabling more effective outbreak response compared to culture-based methods. Clinical presentation of *L. micdadei* infections can differ from other *Legionella* spp. cases and could be missed without a high level of suspicion and access to environmental detection and diagnostic testing. In its current form, the PCR can help public health investigate *L. micdadei* infections and enable heightened surveillance of suspected or confirmed environmental sources. After clinical validation with patient specimens, this PCR could also be used to diagnose patients faster.

As a limitation, this study relied on PCR assays developed and evaluated with specimens collected from a limited number of sources. While a range of environmental matrices (tap water, water faucet swabs, soil and dust) was included, few (soil and mulch) naturally contain *L. micdadei*. Thus, the detection of the target organism from water specimens was confirmed by simulation (i.e., spiking bacteria). This evaluation does not fully represent the complexity and variability encountered in the environment.

## 5. Conclusions

Our *L. micdadei*-specific PCR assay has the potential to provide rapid and accurate detection of a rare opportunistic pathogen, enhancing environmental surveillance. *Legionella* species-specific assays like the one presented in this report can provide public health authorities with a better understanding of environmental sources and transmission dynamics, thus enable better responses to outbreaks. Such advances are required as epidemiological surveillance indicates an increase in cases of Legionnaires’ disease and Pontiac fever [40]. The utility of this PCR assay could be expanded by validating it in respiratory specimens from infected patients, thereby enhancing diagnosis and facilitating targeted therapy, particularly in immunocompromised patients at an increased risk of severe disease.

## Figures and Tables

**Figure 1 idr-17-00131-f001:**
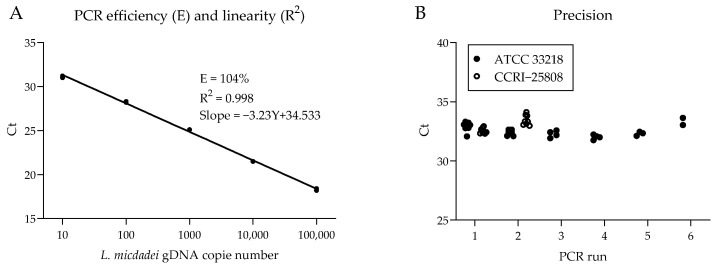
PCR amplification efficiency, repeatability, and reproducibility. (**A**) Standard curve for PCR efficiency and linearity assessment of Ct values against gDNA dilutions of *L. micdadei* ATCC 33218 strain. (**B**) Variability of Ct values across eight independent PCR runs using 10 genome copies of gDNA from *L. micdadei* strains ATCC 33218 and CCRI-25808.

**Figure 2 idr-17-00131-f002:**
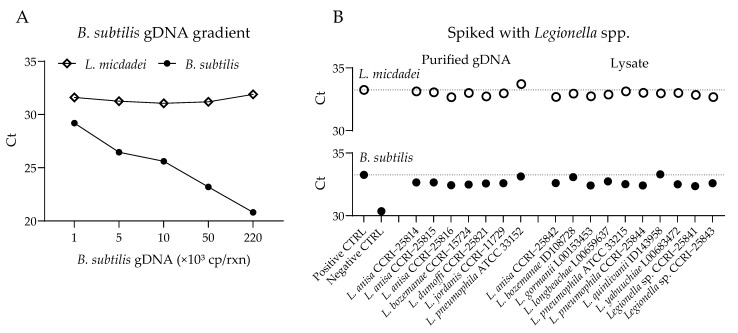
Competition assays for PCR amplification of *L. micdadei* and *B. subtilis* in the presence of *Legionella* spp. (**A**) Amplification with *L. micdadei* (10 genome copies (cp) gDNA per reaction) and increasing amounts of internal control *B. subtilis* gDNA. (**B**) Amplification of *L. micdadei* (10 cp gDNA) and *B. subtilis* (1000 cp gDNA) with the addition of another *Legionella* sp., introduced as either purified gDNA (1 ng) or lysate (1:100 dilution). Seventeen *Legionella* spp. were tested.

**Table 1 idr-17-00131-t001:** PCR results of *Legionella micdadei* and other *Legionella* spp. strains used in this study.

		Isolation			PCR Ct		
Species	Strain	Source	Year	gDNA per rxn	*L. micdadei*	Internal Control	PCR Interp.
*Legionella micdadei*	CCRI-25807	Human	Unk	10 copies	33.10	32.80	D
	CCRI-25808	Human	2024	10 copies	33.10	32.60	D
	CCRI-25809	Human	2024	10 copies	33.70	32.70	D
	CCRI-25810	Human	2024	10 copies	33.50	32.80	D
	CCRI-25817	Human	2024	10 copies	35.00	32.70	D
	CCRI-25819	Human	2023	10 copies	32.20	32.70	D
	CCRI-25820	Human	2023	10 copies	33.00	32.50	D
	ATCC 33218	Human	1980	10 copies	32.20	32.95	D
	ID063037	Human	Unk	Lysate	24.30	-	D
	ID081869	Human	Unk	Lysate	24.70	-	D
	ID088926	Human	Unk	Lysate	25.20	32.42	D
	ID090456	Human	Unk	Lysate	25.10	30.35	D
	ID103059	Human	Unk	Lysate	24.50	-	D
	ID108784	Human	Unk	Lysate	24.80	-	D
	ID114570	Human	Unk	Lysate	25.10	33.40	D
	ID125965	Human	Unk	Lysate	25.30	-	D
	L00132833	Human	Unk	Lysate	25.20	33.60	D
	L00166407	Human	Unk	Lysate	25.10	32.39	D
	L00495253	Human	Unk	Lysate	25.10	33.25	D
	CCRI-25845	Human	2024	Lysate	24.60	-	D
*Legionella anisa*	CCRI-25814	Env	2024	Lysate	ND	32.23	ND
	CCRI-25815	Env	2024	Lysate	ND	32.52	ND
	CCRI-25816	Env	2024	Lysate	ND	32.73	ND
	CCRI-25842	Water	Unk	Lysate	ND	29.04	ND
*Legionella bozemanae*	ATCC 33217	Human	2005	1ng	ND	32.43	ND
	ID108728	Human	Unk	Lysate	ND	29.13	ND
*Legionella dumoffii*	CCRI-25821	Env	2021	Lysate	ND	32.39	ND
*Legionella gormanii*	L00153453	Human	Unk	Lysate	ND	29.24	ND
*Legionella jordanis*	CCRI-11729	River Water	Unk	1ng	ND	32.83	ND
*Legionella longbeachae*	L00659637	Human	Unk	Lysate	ND	29.33	ND
*Legionella pneumophila*	ATCC 33152	Human	Unk	1ng	ND	32.79	ND
	ATCC 33215	Human	Unk	Lysate	ND	32.34	ND
	CCRI-25844	Water	Unk	Lysate	ND	29,26	ND
*Legionella quinlivanii*	ID143958	Human	Unk	Lysate	ND	29.16	ND
*Legionella* sp.	CCRI-25841	Water	Unk	Lysate	ND	29.16	ND
	CCRI-25843	Water	Unk	Lysate	ND	29.24	ND
*Legionella yabuuchiae*	L00683472	Human	Unk	Lysate	ND	29.07	ND

Abbreviations: rxn = reaction; PCR interp. = PCR interpretation; Env = environment; D = detected; ND = not detected; Unk = unknown.

**Table 2 idr-17-00131-t002:** Primer and probe characteristics.

Name	Oligonucleotide	Sequence (5′ → 3′)	Conc^n^ [µM]	Length (nt)	Tm ^1^ (°C)	Amplicon Size (bp)
** *Legionella micdadei* ** **set**						
23S–5S_LmicF	Forward primer	ACTGCCTTTAGGTTATGAGTGA	0.4	22	62.7	205
Pan-Legionella_R ^2^	Reverse primer	TTCACTTCTGAGTTCGAGATGG	0.4	22	63.1	
Lmicdadei-T1-F2 ^2,3^	TaqMan probe	AGCTGATTGGTTAATAGCCCAATCGG	0.2	26	66.9	-
** *Bacillus subtilis* ** **internal control set**						
ABsub150	Forward primer	GCCTCTTCATTTAGGTGATGATAC	0.4	24	62.7	412
ABsub542	Reverse primer	GCCGGCGAATACAGAGATAC	0.4	20	63.1	
Aspores-T1-G2 ^3^	TaqMan probe	ATGGCATCTACAGAYGGTRTTCAGCGC	0.3	27	66.9	-

^1^ Tm calculated using the OligoAnalyzer™ Tool (Integrated DNA Technologies). ^2^ Oligonucleotides for *Legionella* spp. from Cross et al. ^3^ TaqMan probes with a CalRed610 fluorophore at the 5′ end and a BHQ2 quencher at the 3′ end.

**Table 3 idr-17-00131-t003:** Limit of detection of the *L. micdadei* PCR assay.

*L. micdadei* Strains	Positive Amplification per cp Number				
	20 cp	10 cp	5 cp	2 cp	1 cp
ATCC 33218, *N* = 16	16	16	16	16	13
CCRI-25808, *N* = 16	16	16	16	13	9

Abbreviation: cp = genome copy number of *L. micdadei* gDNA.

**Table 4 idr-17-00131-t004:** PCR repeatability and reproducibility for precision evaluation.

*L. micdadei*		Repeatability (Intra-Run)						Reproducibility
Strains ^1^	PCR Runs	1	2	3	4	5	6	(Inter-Run)
ATCC 33218	Ct (avg)	32.9	32.4	32.3	32.0	32.3	33.3	32.5
	CV%	1.2	0.7	0.9	0.6	0.5	1.3	1.4
	n	8	8	4	4	3	2	29
CCRI-25808	Ct (avg)	32.5	33.5	-	-	-	-	33.0
	CV%	0.7	1.3	-	-	-	-	2.0
	n	8	8	-	-	-	-	16

^1^ Input: 10 genome copies of gDNA from *L. micdadei* strains per reaction. Abbreviations: avg = average; CV% = coefficient of variation (%); n = number of replicates.

**Table 5 idr-17-00131-t005:** Detection of *L. micdadei* in environmental specimens.

Specimen Type	Specimen	*L. micdadei* PCR Ct	IC PCR Ct	PCR Interpretation	Culture
Tap water	Neg Control 1	ND	30.28	ND	No growth
Tap water	Neg Control 2	ND	29.93	ND	No growth
Tap water	1	ND	ND	Invalid	*L. pneumophila*
Tap water	1 ^1^	ND	32.72	ND	*L. anisa*
Tap water	2	ND	ND	Invalid	*L. anisa*
Tap water	2 ^1^	ND	31.08	ND	No growth
Tap water	3	ND	30.38	ND	*L. anisa*
Tap water	4	ND	33.26	ND	No growth
Tap water	5	ND	30.15	ND	No growth
Tap water	6	ND	30.21	ND	No growth
Tap water	7	ND	30.15	ND	*L. feeleii*
Tap water	8	ND	30.21	ND	No growth
Swab of faucet	9	ND	30.69	ND	*L. pneumophila*
Swab of faucet	10	ND	29.94	ND	No growth
Dust	11	ND	32.92	ND	N/A
Wood mulch mixed with soil	12	31.68	ND	D	N/A
Gardening soil	13	ND	ND	Invalid	N/A
Gardening soil	13 ^2^	ND	30.21	ND	N/A
Gardening soil	14	ND	ND	Invalid	N/A
Gardening soil	14 ^2^	38.04	30.45	D	N/A
Gardening soil	15	ND	29.20	ND	N/A
Tap water	spiked with *L. micdadei*	29.83	30.33	D	*L. micdadei*
Pos ctrl	10 copies gDNA *L. micdadei*	30.36	29.87	D	N/A
Pos ctrl	10 copies gDNA *L. micdadei*	30.58	29.86	D	N/A
Pos ctrl	10 copies gDNA *L. micdadei*	32.39	32.75	D	N/A
Neg ctrl	TE 1X	ND	29.72	ND	N/A
Neg ctrl	TE 1X	ND	32.14	ND	N/A
Neg ctrl	TE 1X	ND	30.07	ND	N/A

^1^ These specimens inhibited the PCR internal control and were diluted 2/5 for the PCR assay. ^2^ These specimens inhibited the PCR internal control and were diluted 1/5 for the PCR assay. Abbreviations: ND: Not detected. D: Detected. IC: Internal control. N/A: Not applicable.

## Data Availability

All relevant data supporting the findings of this study are included within the article. No additional datasets were generated or analyzed.

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
