# Peer review of "Development of a PCR Assay for the Detection of Legionella micdadei in the Environment"

_2036-7449, 2025, doi:10.3390/idr17050131_

Round 1
Reviewer 1 Report
Comments and Suggestions for Authors
The authors present, in general terms, a study on a bacterium that has a significant impact on some regions of the planet, interacting with human health globally. They identified the underlying problem as the time to diagnosis and the flaws that sometimes occur in the differential component and aim to contribute to the establishment of faster and more accurate methodologies, with the corresponding operational advantages for public health. After a careful reading, I make a set of observations, by area of the article, in an attempt to contribute to its overall improvement, given that it already has a very high level of organization and structure.
Title
The title is fully appropriate for the rest of the work, conveying the necessary and correct information.
Abstract
The abstract is structured, thus allowing a more intuitive overview.
Introduction
The introduction provides good context, presenting the problems identified by the authors in an integrated manner, supported by epidemiological data. I could try to reference some studies (if any?) that are similar to the proposal. Materials and Methods
There is a detailed description, which allows, if necessary, the work to be replicated. I propose justifying the operational advantages for defining the target; were any analyses repeated?
Results
These are supported by three tables and two figures to support the presentation of the data. It seems to me that there is still room for some improvements in data processing, particularly with the implementation of some statistics that would further validate the results the authors proposed to explore.
Discussion
The discussion is well-organized; that is, it discusses the results found and attempts to connect them with other works. However, it seems to me that this is the area with the greatest room for improvement in this article, as it could analyze and discuss more fully the results, their origins, and the benefits and costs, more generally, that these processes may present.
Conclusion
It is fully related to the article and is clearly adequate; I only suggest something that I can reinforce as a complement and not a replacement (for now?)
Therefore, I share these ideas for improvement, leaving their implementation to the authors' consideration. I am available for any clarifications you deem useful.
Author Response
The authors present, in general terms, a study on a bacterium that has a significant impact on some regions of the planet, interacting with human health globally. They identified the underlying problem as the time to diagnosis and the flaws that sometimes occur in the differential component and aim to contribute to the establishment of faster and more accurate methodologies, with the corresponding operational advantages for public health. After a careful reading, I make a set of observations, by area of the article, in an attempt to contribute to its overall improvement, given that it already has a very high level of organization and structure.
Comments 1: Title: The title is fully appropriate for the rest of the work, conveying the necessary and correct information.
Response 1: We would like to thank Reviewer 1 for taking the time to review our manuscript and for the insightful comments provided.
Comments 2: Abstract: The abstract is structured, thus allowing a more intuitive overview.
Response 2: Thank you for your positive feedback regarding the structure of the abstract.
Comments 3: Introduction: The introduction provides good context, presenting the problems identified by the authors in an integrated manner, supported by epidemiological data. I could try to reference some studies (if any?) that are similar to the proposal.
Response 3: In response, we have expanded our introduction to better contextualize the evolution of PCR-based detection methods for Legionella spp., explicitly addressing the limitations faced in environmental matrices, especially for species such as L. micdadei. As recommended, we have added additional literature supporting these claims.
Lines 117-125: “Multiple PCR-based methods have been developed for the detection and identification of Legionella spp., including both genus-level and species-specific assays targeting genetic regions such as 16S rRNA, mip, and various intergenic spacer regions [25–31]. While these approaches have improved diagnostic capabilities, the detection of Legionella spp. like L. micdadei remains challenging, especially in environmental specimens where specificity and sensitivity may be compromised by potential inhibitors and competing bacterial flora [31,32]. Applications that can discriminate between closely related Legionella spp. with high specificity in environmental matrices are limited. In this study, we present a PCR assay that specifically detects L. micdadei [25].”
Comments 4: Materials and Methods: There is a detailed description, which allows, if necessary, the work to be replicated. I propose justifying the operational advantages for defining the target; were any analyses repeated?
Response 4: The PCR assay was designed to target the three genome copies of 23S-5S rRNA intergenic spacer of L. micdadei, thereby increasing the number of detectable templates and improving assay sensitivity when DNA concentrations are low. Additionally, this spacer exhibits highly conserved extremities across the Legionella genus, while its central section varies between species, enabling robust species-specific discrimination within the genus. This combination of high sensitivity and specificity is widely recognized as an optimal strategy in the design of diagnostic PCR assays. We have added this rationale to the Methods section for clarity.
Lines 158-161: “We targeted the three genome copies of 23S-5S rRNA intergenic spacers of L. micdadei, thereby increasing the number of detectable templates and enhancing assay sensitivity [35].”
Comments 5: Results: These are supported by three tables and two figures to support the presentation of the data. It seems to me that there is still room for some improvements in data processing, particularly with the implementation of some statistics that would further validate the results the authors proposed to explore.
Response 5: We appreciate the reviewer’s suggestion regarding the implementation of statistics to further validate the results. To strengthen the manuscript, we have now calculated and reported binomial 95% confidence intervals for specificity and inclusivity, using the exact Clopper–Pearson interval method. We have modified the text accordingly.
Lines 222-225: “Binomial 95% confidence intervals for analytical specificity and inclusivity were calculated using the exact Clopper–Pearson interval method on RStudio (v2025.05.1+513; Posit PBC, Boston, MA) [37].”
Lines 329-336: “The assay showed 100% inclusivity (20/20, 95% CI: 83.2–100%) for L. micdadei strains (Table 1). Analytical specificity was 100% (0/102 false positives; 95% CI: 96.5–100%), with no amplification detected in 17 strains of Legionella spp. other than L. micdadei, 56 bacterial strains from other genera, 6 fungal strains, and 3 human DNA specimens (Suppl. Table 1).
For our limit of detection (LOD) analysis, we initially attempted curve fitting using logistic regression. The data did not provide enough transition points to support a meaningful statistical fit, as nearly all replicates at higher copy numbers were positive and only a sharp drop occurred at the lowest doses. As recommended by clinical and ISO/TS 12869:2019 standards, we therefore reported the empirical LOD: the lowest concentration at which at least 95% of replicates were positive (5 copies/reaction).
Comments 6: Discussion: The discussion is well-organized; that is, it discusses the results found and attempts to connect them with other works. However, it seems to me that this is the area with the greatest room for improvement in this article, as it could analyze and discuss more fully the results, their origins, and the benefits and costs, more generally, that these processes may present.
Response 6: To address this comment about benefits and costs, we have added new detail to the Discussion section:
Lines 430-435: “In addition to these technical strengths, the species-specific PCR assay for L. micdadei of-fers operational advantages for public health and environmental monitoring. The rapid turnaround time (3-4 hours) and low detection threshold facilitate L. micdadei detection and support surveillance of water and soil sources, potentially enabling more effective outbreak response compared to culture-based methods.”
Comments 7: Conclusion: It is fully related to the article and is clearly adequate; I only suggest something that I can reinforce as a complement and not a replacement (for now?)
Response 7: We thank the reviewer for the positive feedback and their careful reading of our conclusion.
Comments 8: Therefore, I share these ideas for improvement, leaving their implementation to the authors' consideration. I am available for any clarifications you deem useful.
Response 8: Thank you for your thoughtful feedback and for taking the time to review the article. We appreciate your suggestions and have carefully considered them during the revision process to improve the manuscript.
Reviewer 2 Report
Comments and Suggestions for Authors< !--StartFragment -->
Introduction
Comment 1: In line 48-49 The sentence “These bacteria can colonize water systems, including cooling towers, hot water tanks, and drinking water systems” is misleading in lines 48–49. While Legionella may be detected in potable water distribution systems, the phrasing "drinking water systems" can imply contamination of the actual drinking water supply. I recommend revising the sentence to “building water distribution systems or water for human consumption” to avoid misinterpretation.
Comment 2: In line 78 -79 PCR-based methods address these gaps by targeting conserved Legionella spp. DNA sequences, delivering results in hours. Please specify if possible how many hours to make the comparison with culture methods clearer.
Materials and Methods
Comment 3: Lines 189-199. I would like more information regarding the methodology for taking water samples, the method and time of transporting the samples to the laboratory, and the use of the ISO methodology. Additionally, please clarify whether sodium thiosulfate was added to the sterile bottles to neutralize residual chlorine during water sampling, as this is standard practice in Legionella culture and PCR protocols.
Discussion
Please explain why no clinical samples were tested even though the assay might ultimately be most helpful for patient diagnosis. L. pneumophila is typically the primary focus of environmental investigations. Since L. micdadei was only found in soil samples and not in water samples, how will this method be helpful? The discussion should also go into detail about the ways in which soil can expose people; for instance, farmers might only be in danger when irrigation techniques produce aerosols or droplets.
Conclusion< !--StartFragment -->
With an emphasis on usefulness and possible future uses, this study showcases innovative developments in environmental surveillance. Although clinical applications—specifically in respiratory specimens and therapy—are discussed as potential future developments, more evidence is needed to establish a direct connection between Pontiac fever and climate change.
General comment: Please ensure consistent taxonomy and formatting. Species names should be italicised, genus names should be shortened after initial mention (e.g., L. micdadei), and Legionella spp. should be used correctly versus Legionella sp.
< !--StartFragment -->
Author Response
Introduction
Comments 1: In line 48-49 The sentence “These bacteria can colonize water systems, including cooling towers, hot water tanks, and drinking water systems” is misleading in lines 48–49. While Legionella may be detected in potable water distribution systems, the phrasing "drinking water systems" can imply contamination of the actual drinking water supply. I recommend revising the sentence to “building water distribution systems or water for human consumption” to avoid misinterpretation.
Response 1: Thank you for this helpful suggestion. We agree that the use of “drinking water systems” could be misinterpreted. We have revised the sentence accordingly.
Lines 71-72: “They can colonize water systems, including cooling towers, hot water tanks, and water distribution systems, posing significant public health risks [3].”
Comments 2: In line 78 -79 PCR-based methods address these gaps by targeting conserved Legionella spp. DNA sequences, delivering results in hours. Please specify if possible, how many hours to make the comparison with culture methods clearer.
Response 2: A time range has been added for clarity. Literature shows that PCR-based methods can deliver Legionella spp. testing results in approximately 6 to 8 hours.
Lines 111-113: “Polymerase chain reaction (PCR)-based methods address these limitations by targeting conserved Legionella spp. DNA sequences, delivering results within 6 to 8 hours [19–21].”
In the discussion, we also clarified our sample preparation and PCR turn-around time. See response to comment 6 of reviewer 1.
Materials and Methods
Comments 3: Lines 189-199. I would like more information regarding the methodology for taking water samples, the method and time of transporting the samples to the laboratory, and the use of the ISO methodology. Additionally, please clarify whether sodium thiosulfate was added to the sterile bottles to neutralize residual chlorine during water sampling, as this is standard practice in Legionella culture and PCR protocols.
Response 3: We have expanded the Methods section to include the following details:
Lines 274-277: “One-liter tap water specimens were collected from hospital faucets in sterile bottles and processed on the same day, following ISO 12869:2019 guidelines. However, sodium thiosulfate was not added to the bottles to neutralize chlorine.”
Discussion
Comment 4: Please explain why no clinical samples were tested even though the assay might ultimately be most helpful for patient diagnosis. L. pneumophila is typically the primary focus of environmental investigations. Since L. micdadei was only found in soil samples and not in water samples, how will this method be helpful? The discussion should also go into detail about the ways in which soil can expose people; for instance, farmers might only be in danger when irrigation techniques produce aerosols or droplets.
Response 4: Thank you for your comment. Clinical respiratory specimens were not part of the PCR evaluation in this first phase of the project, which focused on environmental specimens where multiple Legionella spp. could cohabitate. Our focus was on method development and evaluation using cultured strains and environmental matrices. We investigated the presence of the L. micdadei from different environments (indoor, outdoor, water, soil), including diverse specimens. We have addressed your comment about clinical relevance by adding the following discussion in the revised manuscript:
In the introduction, we clarified further the presence of Legionella spp in the soil.
Lines 72-76. “In addition, certain Legionella spp. have also been detected in soils, potting mixes, and composts. These can serve as additional environmental reservoirs and sources of infection, particularly for individuals exposed to contaminated dust or aerosols during gardening or agricultural work [4–6].”
Lines 396-399: “Exposure to soil contaminated by Legionella spp. can occur through aerosolization during soil-disturbing activities and handling [37–39]. Thus, soil testing is important for outbreak investigation and surveillance.”
Conclusion
Comments 5: With an emphasis on usefulness and possible future uses, this study showcases innovative developments in environmental surveillance. Although clinical applications—specifically in respiratory specimens and therapy—are discussed as potential future developments, more evidence is needed to establish a direct connection between Pontiac fever and climate change.
Response 5: Thank you for your comment. The manuscript does not claim that this study establishes a direct connection between Pontiac fever and climate change. The text has been revised accordingly.
Lines 459-462: “Such advances are required as epidemiological surveillance indicates an increase in cases of Legionnaires’ disease and Pontiac fever [41].”
Comments 6: General comment: Please ensure consistent taxonomy and formatting. Species names should be italicised, genus names should be shortened after initial mention (e.g., L. micdadei), and Legionella spp. should be used correctly versus Legionella sp.
Response 6: Thank you for your comment. We used the abbreviated form L. micdadei after defining the genus and species in the text. For Table 1, we kept the genus and species nomenclature to maintain symmetry, as other Legionella spp. were not previously defined. We then only use abbreviated nomenclatures. The manuscript has been carefully reviewed to ensure consistent taxonomy and formatting.
Reviewer 3 Report
Comments and Suggestions for Authors
This paper reports the development and basic performance of a species-specific PCR assay for the environmental detection of Legionella micdadei. This study satisfies the need and is particularly significant in its application to environmental matrices. However, improvements are essential in the logical development of the paper, experimental design, interpretation, sufficiency of the results, and practical applicability.
- The study primarily relies on spiking experiments in water and a small number of soil samples, without testing naturally L. micdadei-positive environmental specimens. As a result, validation of detection performance in real-world conditions is lacking. Since most tap water, swabs, and other materials examined did not naturally contain L. micdadei but were artificially spiked, the assay’s practical sensitivity and robustness remain uncertain. Therefore, the conclusions regarding its utility for environmental surveillance should be tempered, and these limitations more explicitly acknowledged.
- Compared with previously reported PCR assays targeting similar regions, the unique contribution of this assay specifically the added advantages of the chosen primer/probe combination is not clearly demonstrated. Moreover, the negative controls seems limited. How does the assay’s sensitivity, specificity, and limit of detection compare with previously reported assays for L. micdadei or pan-Legionella PCR?
- The Methods section is overly long and contains considerable repetition. I recommend summarizing key information—such as sample numbers and distribution, experimental workflows in tables or schematic figures, while providing a more concise narrative in the main text. This would improve clarity and readability without sacrificing essential detail.
- Please be sure to spell out some abbreviations when they are first used in the text.
- Although specificity was evaluated against multiple Legionella spp. and other bacteria, the discussion should more explicitly address the potential risk of cross-reactivity, especially considering that the reverse primer shows partial sequence conservation across the genus.
- While the introduction and discussion emphasize the clinical challenges of diagnosing L. micdadei pneumonia, the study itself focuses exclusively on environmental applications. The authors should elaborate on how and under what circumstances this assay could be translated into clinical diagnostics, and address potential regulatory or methodological hurdles that may need to be overcome.
- The manuscript would benefit from professional language editing to improve clarity and flow.
- The inclusion of an Informed Consent Statement appears unnecessary, as the study did not involve human participants or patient-derived specimens. The authors should clarify this point.
Author Response
This paper reports the development and basic performance of a species-specific PCR assay for the environmental detection of Legionella micdadei. This study satisfies the need and is particularly significant in its application to environmental matrices. However, improvements are essential in the logical development of the paper, experimental design, interpretation, sufficiency of the results, and practical applicability.
Comments 1: The study primarily relies on spiking experiments in water and a small number of soil samples, without testing naturally L. micdadei-positive environmental specimens. As a result, validation of detection performance in real-world conditions is lacking. Since most tap water, swabs, and other materials examined did not naturally contain L. micdadei but were artificially spiked, the assay’s practical sensitivity and robustness remain uncertain. Therefore, the conclusions regarding its utility for environmental surveillance should be tempered, and these limitations more explicitly acknowledged.
Response 1: Thank you for carefully reviewing the manuscript and providing these constructive comments. The Discussion section has been revised to clarify that evaluation was performed using spiked tap water, water faucet swab, dust, and soil samples, but that most samples did not naturally contain L. micdadei. While these results demonstrate sensitivity and specificity in controlled settings, L. micdadei is not commonly found in the environment in Quebec, where the study was performed. Finding two different sources of L. micdadei in the environment was important. The text now emphasizes the necessity for additional real-world validation and advises that conclusions regarding environmental surveillance utility should be interpreted in light of this limitation.
Lines 441-446: “As a limitation, this study relied on PCR assays developed and evaluated with specimens collected from a limited number of specimens. While a range of environmental matrices (tap water, water faucet swabs, soil and dust) was included, few (soil and mulch) naturally contain L. micdadei. Thus, the detection of the target organism from water specimens was confirmed by simulation (i.e. spiking bacteria). This evaluation does not fully represent the complexity and variability encountered in the environment.”
Comments 2: Compared with previously reported PCR assays targeting similar regions, the unique contribution of this assay specifically the added advantages of the chosen primer/probe combination is not clearly demonstrated. Moreover, the negative controls seems limited. How does the assay’s sensitivity, specificity, and limit of detection compare with previously reported assays for L. micdadei or pan-Legionella PCR?
Response 2: The revised manuscript now clarifies that our assay achieves a lower limit of detection (5 genome copies per reaction at ≥95% detection) compared to the prior assays by Cross et al. (2016), which achieved 50% detection at 10 copies. We also now note that, uniquely, our method includes an internal amplification control to detect the presence of PCR inhibitors or failed reactions. These revisions are now included in the Discussion section.
Lines 419-429: “The present L. micdadei-specific PCR assay offers distinct advantages over previously published studies[25]. The Cross et al. assay detected L. micdadei at 10 genome copies per reaction with 50% positivity. Our assay demonstrated improved sensitivity with a lower LOD of 5 genome copies per reaction in ≥95% of replicates. Environmental matrices frequently contain substances like humic acids, which may co-extract with DNA and act as potent PCR inhibitors, interfering with nucleic acid amplification and potentially leading to false negatives or reduced assay sensitivity [39,40]. Thus, we incorporated an internal amplification control (B. subtilis DNA) to monitor that each PCR reaction worked correctly. This internal control is especially important to detect false negatives when testing environmental specimens containing PCR inhibitors.”
Comments 3: The Methods section is overly long and contains considerable repetition. I recommend summarizing key information—such as sample numbers and distribution, experimental workflows in tables or schematic figures, while providing a more concise narrative in the main text. This would improve clarity and readability without sacrificing essential detail.
Response 3: Thank you for the thoughtful suggestion regarding streamlining the Methods section by incorporating summary visuals and narrative condensation. The methods section has been revised for clarity and conciseness, reducing the word count from 1,453 to 1,281 words.
Comments 4: Please be sure to spell out some abbreviations when they are first used in the text.
Response 4: All abbreviations are now spelled out at their first mention in the revised text.
Comment 5: Although specificity was evaluated against multiple Legionella spp. and other bacteria, the discussion should more explicitly address the potential risk of cross-reactivity, especially considering that the reverse primer shows partial sequence conservation across the genus.
Response 5: Thank you for this observation. We have improved the discussion accordingly.
Lines 429-430 “Finally, our assay tolerates competition with more than 1,000 fold excess concentration of other Legionella spp. strains [35].”
Comments 6: While the introduction and discussion emphasize the clinical challenges of diagnosing L. micdadei pneumonia, the study itself focuses exclusively on environmental applications. The authors should elaborate on how and under what circumstances this assay could be translated into clinical diagnostics, and address potential regulatory or methodological hurdles that may need to be overcome.
Response 6: Thank you for highlighting this point. The present study evaluates the assay exclusively for environmental applications, as detection of L. micdadei is highly valuable for epidemiological investigations and surveillance. The L. micdadei PCR assay was designed to be specific against a comprehensive panel of respiratory pathogens and human DNAs. As shown in supplemental Table 1, many respiratory pathogens were tested without cross-reactivity. Full clinical implementation would require laboratory validation on patient specimens positive for L. micdadei, other Legionella spp. and other respiratory pathogens following Clinical Laboratory Standards Institute guidelines.
Lines 435-440: “Clinical presentation of L. micdadei infections can differ from other Legionella spp. cases and could be missed without a high level of suspicion and access to environmental detection and diagnostic testing. In its current form, the PCR can help public health investigate L. micdadei infection and enable heightened surveillance of suspected or confirmed environmental sources. After clinical validation with patient specimens, this PCR could also be used to diagnose patients faster.”
Comments 7: The manuscript would benefit from professional language editing to improve clarity and flow.
Response 7: Thank you for your suggestion. The manuscript will be carefully reviewed and edited for clarity and language.
Comments 8: The inclusion of an Informed Consent Statement appears unnecessary, as the study did not involve human participants or patient-derived specimens. The authors should clarify this point.
Response 8: We wanted to ensure that contamination from human DNA would not interfere with PCR testing, as the environmental specimens are collected by humans and to anticipate future clinical implementation. Human DNA was used as part of the analytical specificity testing, with three human DNA specimens included to evaluate the assay. Use of DNA for research required ethics approval from our institution, as mentioned in the manuscript.
Round 2
Reviewer 1 Report
Comments and Suggestions for Authors
After rereading both the document with the responses to my comments and, especially, the article in its current version, I thank the authors for incorporating most of my suggestions (and, from what I understand from comparing the initial version, they certainly also made changes suggested by other reviewers). There is a clear improvement in the article overall, with the scientific explanation and the way the message is conveyed now clearer and more direct. It seems to me that the discussion, while adding benefits, contains little mention of costs or limitations, certainly has to do with the article's strategy.
At this stage, I have nothing further to add, leaving the final decision to the Editor. I thank the authors and wish them well, both with this article and in future work.
Reviewer 3 Report
Comments and Suggestions for Authors
I have confirmed.